# Late-Onset Sepsis Mortality among Preterm Infants: Beyond Time to First Antibiotics

**DOI:** 10.3390/microorganisms11020396

**Published:** 2023-02-03

**Authors:** Francesca Miselli, Sara Crestani, Melissa Maugeri, Erica Passini, Valentina Spaggiari, Elisa Deonette, Branislava Ćosić, Katia Rossi, Maria Federica Roversi, Luca Bedetti, Licia Lugli, Riccardo Cuoghi Costantini, Alberto Berardi

**Affiliations:** 1PhD Program in Clinical and Experimental Medicine, University of Modena and Reggio Emilia, 41121 Modena, Italy; 2Neonatal Intensive Care Unit, University Hospital of Modena, 41124 Modena, Italy; 3Pediatric Post-Graduate School, University Hospital of Modena and Reggio Emilia, 41224 Modena, Italy; 4Medicine and Surgery School, University of Modena and Reggio Emilia, 41124 Modena, Italy; 5Department of Medical and Surgical Sciences for Mother, Child and Adult, University of Modena and Reggio Emilia, 41124 Modena, Italy

**Keywords:** late-onset sepsis, neonatal sepsis, empirical antimicrobials, time to antibiotics, first antibiotics, effective antimicrobials, blood-brain barrier, meningitis, mortality, brain sequelae

## Abstract

**Objective:** To investigate the impact of timing, in vitro activity and appropriateness of empirical antimicrobials on the outcome of late-onset sepsis among preterm very low birth weight infants that are at high risk of developing meningitis. **Study design:** This retrospective study included 83 LOS episodes in 73 very low birth weight infants born at ≤32 weeks’ gestation with positive blood and/or cerebrospinal fluid culture or polymerase chain reaction at >72 h of age. To define the appropriateness of empirical antimicrobials we considered both their in vitro activity and their ideal delivery through the blood-brain barrier when meningitis was confirmed or not ruled out through a lumbar puncture. The primary outcome was sepsis-related mortality. The secondary outcome was the development of brain lesions. Timing, in vitro activity and appropriateness of empirical antimicrobials, were compared between fatal and non-fatal episodes. Uni- and multi-variable analyses were carried out for the primary outcome. **Results:** Time to antibiotics and in vitro activity of empirical antimicrobials were similar between fatal and non-fatal cases. By contrast, empirical antimicrobials were appropriate in a lower proportion of fatal episodes of late-onset sepsis (4/17, 24%) compared to non-fatal episodes (39/66, 59%). After adjusting for Gram-negative vs. Gram-positive pathogen and for other supportive measures (time to volume administration), inappropriate empirical antimicrobials remained associated with mortality (aOR, 10.3; 95% CI, 1.4–76.8, *p* = 0.023), while timing to first antibiotics was not (aOR 0.9; 95% CI, 0.7–1.2, *p* = 0.408; AUC = 0.88). The association between appropriate antimicrobials and brain sequelae was also significant (*p* = 0.024). **Conclusions:** The risk of sepsis-related mortality and brain sequelae in preterm very low birth weight infants is significantly associated with the appropriateness (rather than the timing and the in vitro activity) of empirical antimicrobials. Until meningitis is ruled out through lumbar puncture, septic very low birth weight infants at high risk of mortality should receive empiric antimicrobials with high delivery through the blood-brain barrier.

## 1. Introduction

Worldwide, sepsis is still a major cause of neonatal morbidity and mortality, although precise estimates of its burden vary by setting, particularly between countries of different income levels. Late-onset sepsis presents beyond 3 days of age and is caused by microorganisms acquired from the community or the hospital environment. In very low birth weight infants (birth weight ≤ 1500 g) and high-risk preterm infants, for whom the hospital stay after birth is prolonged, the definition of late-onset sepsis refers to any episode of sepsis occurring from 3 days of life to hospital discharge [1] and its related morbidity and mortality are high [2,3,4,5,6]. In fact, during the first 90 days of life, the innate immune system (namely, phagocytes, natural killer cells, antigen-presenting cells, and the complement system) provide a defence against pathogens, which is weaker among very low birth weight infants [7]. Decreased function of neutrophils and low concentrations of immunoglobulins increase the susceptibility of preterm infants to invasive infections, increasing their related morbidity and mortality [8]. Hospitalised preterm infants are exposed to environmental organisms that might become pathogenic for their immature immune systems. Contact with hospital staff, family members and contaminated equipment all represent opportunities for pathogen exposure [9]. Hand contamination appears as the most frequent source of postnatal infections among hospitalised very low birth weight infants. In fact, the risk of late-onset sepsis and its related mortality are up to > 50- and > 400-fold higher among very low birth weight as compared to full-term infants [10]. In addition, meningitis may frequently complicate neonatal sepsis. However, the diagnosis of meningitis in infants is challenging for neonatologists because its signs and symptoms are vague and overlap with those of sepsis. Specific signs of involvement of the central nervous system are present in only one-third of cases and vary depending on birth weight and gestational age at birth; in particular, they are even rarer among neonates of lower gestational age [11].

Early recognition and treatment of sepsis are essential for the survival of patients, and the Surviving Sepsis Campaign places these goals at the centre of efforts to improve patient outcomes [12]. Indeed, survival is reduced after delayed, compared to early, antibiotic treatment in children and adults with sepsis [13,14,15]. In contrast, research data do not irrefutably confirm this association in neonates [16], and only a few, recent studies suggest that timely administration of antibiotics improves survival [17,18]. However, aggressive time-to-antibiotic administration may lead to unnecessary antibiotic exposure, with potential adverse consequences on preterm very low birth weight infants’ health. The effect of antibiotic timing and particularly the consequences of delayed antibiotic administration in neonatal sepsis should be investigated in detail, to assess the best risk/benefit ratio in the timing of antimicrobials administration [19].

Notably, the association between survival and timely antibiotic administration may be biased by multiple factors, including the timeliness of diagnosis, the implementation of appropriate shock supportive measures, and, most of all, the effectiveness of the empiric antimicrobial treatment [19]. In the neonatal period, an additional variable should be considered: since meningitis often complicates neonatal sepsis (particularly among preterm infants) [8], in order to be effective, empirical antimicrobials should combine in vitro activity against the pathogen and high delivery through the blood-brain barrier when meningitis is confirmed. However, the lumbar puncture, necessary to rule out meningitis, is often postponed (or not performed) in infants with hemodynamic instability [10,20,21,22]. As a result, often the empirical antimicrobials’ appropriateness remains undetermined in neonatal sepsis.

In this study, we assessed the appropriateness of empirical antimicrobials in neonatal sepsis, based on their in vitro activity and the presence of meningitis. Thus, we sought to determine the impact of the timing and the appropriateness of empirical antimicrobials on neonatal outcome (namely sepsis-related death and brain sequelae). Our study was focused on preterm very low birth weight infants who are at the highest risk of sepsis-related death and sequelae.

## 2. Materials and Methods

### 2.1. Study Design

We performed a retrospective observational study of episodes of LOS among very low birth weight infants, over a 12-year period (from 1 June 2010 to 1 April 2022). The study was carried out at the Neonatal Intensive Care Unit (NICU) of the University Hospital of Modena, Italy. This is a high-volume level-three facility, with inborn neonates accounting for most admissions. The NICU contains 20 cots, receives approximately 450 admissions per year, and the medical staff consists of 12 physicians. The Institutional Research Ethics Board approved the study (prot. 978/2019). Given the impossibility of retrospectively retrieving the consent for all the infants included in the study, the research ethics committee waived the need for the consent.

### 2.2. Definitions

Late-onset sepsis: positive blood or cerebrospinal fluid culture (or cerebrospinal fluid positive polymerase chain reaction), in an infant (aged 3 to 90 days) with signs and symptoms of sepsis.Signs and symptoms of sepsis: fever, hypothermia or temperature instability, tachycardia, bradycardia or heart rate instability, new or more frequent apnoeas, capillary refill time >2 s, changes in skin colour, increased respiratory support, unstable clinical condition and apathy [23].Time of onset of late-onset sepsis: timing of first signs and symptoms of sepsis reported in electronic medical records.Time to first antibiotics: time interval between the onset of late-onset sepsis and the first administration of empirical antimicrobials.Sepsis-associated mortality: death occurring within 7 days from its onset or clearly related to complications due to late-onset sepsis [24].

In vitro activity of empirical antimicrobials

Active empirical antimicrobials: the pathogen was susceptible (in vitro) to at least one empirical antimicrobial (or each pathogen was susceptible to at least one antimicrobial in the case of polymicrobial infections).Inactive empirical antimicrobials: the pathogen was resistant to all empirical antimicrobials.

Appropriateness of empirical antimicrobials

Appropriate empirical antimicrobials: the empirical antimicrobial was active (in vitro) against the pathogen and (i) meningitis was ruled out (through a lumbar puncture) or (ii) the antimicrobial ideally penetrated at high levels the blood-brain barrier.Inappropriate empirical antimicrobials: (i) inactive empirical antimicrobials or (ii) active empirical antimicrobials ideally did not pass the blood-brain barrier at high levels (e.g., an aminoglycoside) and lumbar puncture findings were consistent with meningitis.Undetermined appropriateness of empirical antimicrobials: active empirical antimicrobials ideally did not pass the blood-brain barrier at high levels (e.g., an aminoglycoside) and meningitis was not ruled out (no lumbar puncture was performed).

### 2.3. Inclusion Criteria

We included all episodes of late-onset sepsis among very low birth weight infants delivered at ≤32 weeks gestation. If infants presented with multiple late-onset sepsis episodes during their hospital stay, all episodes separated by a ≥7 days in-between period and caused by different pathogens were included for the analysis.

### 2.4. Data Collection

The microbiological laboratory database was searched for isolates grown in blood or cerebrospinal fluid cultures and for any positive cerebrospinal fluid polymerase chain reaction during the study period. For each case of late-onset sepsis identified in the laboratory database, study staff fulfilled a standardised form, by collecting data from patients’ electronic medical records (Metavision Electronic Medical Record, iMDsoft, Tel Aviv, Israel). Information was strictly anonymous and included each infant’s baseline characteristics, clinical symptoms, diagnostic evaluation, infecting organisms, management and outcome. The electronic medical records available in our NICU allow for very reliable information on the timing of the collection of blood samples, patients’ clinical worsening and antimicrobials administration. To determine as accurately as possible the timing of the onset of sepsis, electronic medical records were searched for the earliest signs of sepsis [23]. Time to blood culture, time to first antibiotics and time to volume administration were then calculated.

### 2.5. Exclusion Criteria

We excluded episodes of culture-negative late-onset sepsis, as this is a heterogeneous group containing both infants with an undetected bacterial infection and infants who are not truly infected. Coagulase-Negative Staphylococci (CoNS) infections were also excluded because often it is not feasible to distinguish true infection from potential contamination [25,26,27,28,29]. Furthermore, Micrococci, Aerococcus species, Corynebacterium species, Propionibacterium species, Bifidobacterium species, and Bacillus species grown in a single culture were considered contaminants and were excluded from the analysis. Additional exclusion criteria were: gestational age over 32 weeks, major congenital malformations, chromosomal disorders and inherited metabolic diseases.

### 2.6. Outcomes

The primary outcome for this study was sepsis-associated mortality. The secondary outcome was the development of brain sequelae, including intraventricular haemorrhage grade III and IV according to the Papile classification scheme [30], other parenchymal haemorrhages and white matter lesions.

### 2.7. Study Setting

The decision to evaluate and manage sepsis was at the discretion of the attending physician. The standard diagnostic evaluation for late-onset sepsis included the collection of a blood sample (minimum 0.5–1.0 mL) for culture, complete blood count and gas analysis. Lumbar puncture was performed prior to the initiation of the empirical antimicrobials when the infant’s clinical conditions allowed. Additional laboratory tests at the onset and subsequent laboratory monitoring were at the discretion of the attending physician, depending on the preceding test findings and the patient’s clinical condition. According to our local protocol of antimicrobial stewardship [31], broad-spectrum antibiotics (including oxacillin plus an aminoglycoside) were given and discontinued within 48 h if sepsis was ruled out. Empiric antibiotic treatment was not necessarily given after the collection of a blood culture. This decision was usually left to attending the physician, depending on the patient’s clinical condition and, in less severe cases, on the results of the sepsis workup.

### 2.8. Statistical Analysis

Descriptive data are presented as frequencies and percentages for categorical variables, and median, range and IQR for continuous variables. Wilcoxon rank-sum and Pearson’s χ^2^ tests were used to compare continuous and categorical variables, as appropriate. Mortality was the primary outcome of interest. We used logistic regression models to determine the association between management and sepsis-related mortality. Gram-positive vs. Gram-negative pathogen was included as a biological variable in analysis owing to association with outcome. Results from the logistic regression models were reported as odds ratios (OR), relative 95% confidence (95% CI) intervals and *p*-values. A 2-sided *p*-value < 0.05 was considered statistically significant. The data management and all analyses were performed using R 4.2.1.

## 3. Results

During the study period, 83 episodes of late-onset sepsis met inclusion criteria: positive blood culture (*n* = 77, of whom 7 had also a positive cerebrospinal fluid culture); positive cerebrospinal fluid and negative blood culture (*n* = 5); negative blood and cerebrospinal fluid cultures, positive cerebrospinal fluid polymerase chain reaction (*n* = 1). These 83 episodes of late-onset sepsis occurred in 73 infants (6 infants experienced 2, and 2 infants experienced 3 episodes, of late-onset sepsis). All infants with sterile blood culture and cerebrospinal fluid culture positivity were experiencing their first infectious episode. Overall, Gram-negative organisms were yielded in 43 (51.8%) out of 83 episodes of late-onset sepsis (*Enterobacteriaceae n* = 28, *E. coli n* = 9, *P. aeruginosa n* = 4, *Acinetobacter baumanii*, *n* = 1, *Stenotrophomonas maltophilia*, *n* = 1); Gram-positive organisms were yielded in 28 (33.7%) episodes of late-onset sepsis (*S. aureus n* = 11, group B *Streptococcus*, *n* = 11, *Enterococcus n* = 6). Fungi were yielded in 3 (3.6%) episodes (*Candida albicans n* = 2, *Candida parapsilosis n* = 1). Polymicrobial infections (*n* = 9, 10.8%) accounted for the remaining episodes of late-onset sepsis.

Empirical antimicrobials were in vitro active in most episodes of late-onset sepsis (59/83, 71.1%). However, they were appropriate only in approximately half of the cases (43/83, 51.8%); in the remaining 40 cases, antimicrobials were either inappropriate (27/83, 32.5%) or their appropriateness was undetermined (13/83, 15.7%).

Among 83 episodes of late-onset sepsis, 17 (20.5%) were fatal (with death occurring within 7 days from the onset in 14 cases). Among non-fatal episodes (*n* = 66), five (7.6%) developed brain sequelae (two intraventricular haemorrhage, one intraventricular haemorrhage and white matter lesions, one intraventricular haemorrhage and cerebellar haemorrhage, one cerebellar haemorrhage). Demographic and antimicrobials data of fatal versus non-fatal episodes are compared in Table 1. Figure 1 compares the in vitro activity and appropriateness of empirical antimicrobials for non-fatal versus fatal.

When empirical antimicrobials were in vitro inactive, sepsis-related mortality was not significantly higher (OR: 2.778; 95% CI 0.918–8.397; *p* = 0.070). By contrast, the risk of sepsis-related mortality was 4-fold higher when empirical antimicrobials were inappropriate (OR 4.105; 95% CI 1.097–15.359; *p* = 0.036). The risk of sepsis-related mortality was up to 6-fold higher also when the appropriateness of empirical antimicrobials was undetermined (OR 6.093; 95% CI 1.334–27.831; *p* = 0.020). The association between appropriate antimicrobials and brain sequelae was also significant (*p* = 0.024): among cases of late-onset sepsis with brain sequelae (*n* = 5), none had received appropriate empirical antimicrobials (inappropriate, *n* = 4; undetermined, *n* = 1).

The median time to blood culture was 0 h (IQR 0–0.75 h, range 0–49 h). Lumbar puncture results were available in approximately half of the episodes (43/83, 51.8%); in the remaining cases, lumbar puncture was either bloody (and therefore cerebrospinal fluid was not sent to the laboratory for analysis) or not performed. The median time to first antibiotic administration was 2 h (IQR 0–6, range 0–68 h). Time to first antibiotics was significantly different between fatal versus non-fatal cases; however, no effect was detected when analysing how sepsis-related mortality varied by time to first antibiotics (OR: 0.849; 95% CI 0.703–1.024; *p* = 0.087). Time to first antibiotics was also not associated with brain sequelae (OR: 0.999; 95% CI 0.927–1.077; *p* = 0.987).

Fluids were administered in most episodes of late-onset-sepsis (51/83, 61.4%; of which 38/51 were crystalloids). The mean time to volume administration was 2 h (IQR 0.38–9.25, range 0–59 h). The time to volume administration was not associated with sepsis-related mortality (OR 0.922; 95% CI 0.831–1.024; *p* = 0.129). Approximately half of the late-onset sepsis episodes also required inotropes (46/83, 55.4%).

The risk of sepsis-related mortality was higher among Gram-negative infections. The effect of interventions on sepsis-related mortality at uni- and multi-variate analysis are shown in Table 2. In the multivariate model, the appropriateness of empirical antimicrobials remained associated with a higher risk of sepsis-related mortality after adjusting for the type of pathogen (Gram-positive vs. Gram-negative) (area under ROC curve = 0.88, Hosmer–Lemeshow goodness of fit test *p* = 0.301; sensitivity 0.77, specificity 0.90) (Table 3 and Figure 2).

## 4. Discussion

Over the past few years, the need for prompt recognition and treatment of sepsis has become a major objective of quality improvement, exemplified by efforts to raise awareness and implement measures to increase survival, as in the Surviving Sepsis Campaign [12,14]. In our study, time to first antibiotics was not associated with survival rates among preterm very low birth weight infants with late-onset sepsis. We also found no association between time to first antibiotics and brain sequelae. This is not surprising, since in non-critical infants antibiotics were not given at the time of blood culture collection, and sometimes the time interval was delayed by many hours. Our data are consistent with a retrospective study including 142 preterm neonates (gestational age under 35 weeks gestation) with late-onset sepsis. Investigators excluded cases of CoNS, and time to antibiotic administration was comparable between neonates who died and neonates who survived (1.8 vs. 1.9 h respectively, *p* = 0.74) [16]. These results are not consistent with two recent studies demonstrating the protective role of early versus delayed antimicrobials administration in neonatal sepsis (higher survival rates, shorter length of hospital stays, and lower risk of complications) [17,18]. However, both studies had several potential limitations and biases [32]. Firstly, they included both full-term and preterm infants, while our study focused on preterm very low birth weight infants, who have much higher risks of sepsis-related mortality [10]. Secondly, in those cohorts, the most commonly identified pathogens were CoNS, which are not always associated with true infections. Furthermore, the proportion of fatal late-onset sepsis cases due to CoNS is frequently reported as very low in neonates (1%) [33], thus the effect of timely administration of empirical antibiotics on mortality may have been biased. For these reasons, we excluded CoNS infections from our cohort. Third, the authors measured time to first antibiotics as the time interval between the order of blood culture (or antimicrobials, whichever came first) and the actual administration of antimicrobials. However, the timing of the prescription may be delayed with respect to the onset of symptoms of late-onset sepsis (in our cohort such interval was up to 2 days). Thus, in these studies the time to first antibiotic administration could underestimate the time interval from the actual first signs of late-onset sepsis. By contrast, we calculated the time to first antibiotics from the onset of sepsis, retrospectively searching health medical records for the first available signs and symptoms. Finally, studies supporting the association between delayed administration and poor neonatal outcome did not consider some relevant supportive measures (such as volume administration) [32], which instead were included in our analysis.

In the current study, the risk of sepsis-related mortality in preterm very low birth weight infants was significantly associated with the appropriateness (rather than the timing) of empirical antimicrobials. Previous studies have already demonstrated the protective role of effective antibiotics for adults and children with sepsis. An increased survival was found in adults with septic shock when effective antimicrobials were administered within the first hour of documented hypotension [34]. Furthermore, in children with severe sepsis or septic shock (median 7 years of age), an escalating risk of mortality was observed with each hour delay from sepsis recognition to first effective antimicrobial administration, although this did not reach significance until 3 h [13]. However, in both studies, the effectiveness of antimicrobials was limited to their in vitro activity against the pathogen. Delivery of antimicrobials through the blood-brain barrier was not considered. Nevertheless, among neonates, the risk of meningitis is higher as compared to adults and children, since up to 30% of cases of neonatal sepsis may complicate with meningitis, particularly among preterm infants [35,36,37,38]. In fact, newborns’ humoral and cellular immunity and phagocytes’ function are immature. Compared to full-term infants, the transplacental passage of maternal immunoglobulins is substantially lower in infants younger than 32 weeks gestation, and the inefficiency in their alternative complement pathway compromises defences against encapsulated bacteria; T-cell defence and mediation of B-cell activity are also compromised. Finally, deficient migration and phagocytosis by neutrophils contribute to the increased vulnerability even to pathogens of low virulence [7].

Furthermore, the diagnosis of meningitis among newborns is more challenging compared to older children and adults because its clinical presentation is frequently vague and lacks specificity, particularly at symptoms onset. Cerebrospinal fluid culture and analysis remain essential for the diagnosis and lumbar puncture is recommended in guidelines as the gold standard for confirming the diagnosis [39]. Evidence on the risks of the procedure at younger ages is limited. Most contraindications (haemodynamic instability, severe respiratory distress; increased intracranial pressure, disorders of coagulation, skin infection on the site of the tap, abnormal anatomy of the spinal cord and spinal epidural abscess) and complications of lumbar puncture (headache, local pain at the puncture site; rare: epidural, subdural or subarachnoid haemorrhage, osteomyelitis, epidural abscess or discitis, bleeding, brain herniation) come from studies carried out in older children; limited data exist regarding neonates and infants [22]. In a large study, the risk of death was comparable between very low birth weight infants who underwent lumbar puncture (284 of 2877) and those who did not (661 of 6764, approximately 10% in both groups). By contrast, infants with late-onset meningitis were more likely to die compared with those who had a lumbar puncture performed but did not have meningitis (23% versus 9%, *p* = 0.001) [37]. However, in clinical practice, clinicians are often reluctant to perform a lumbar puncture, sometimes deferring (or omitting) the procedure because of concerns regarding potential peri-procedural adverse events (namely, hypoxia or bradycardia) or other complications (traumatic and unsuccessful puncture, brain herniation). Thus, meningitis may remain underdiagnosed among septic infants, complicating with high incidences of death and brain sequelae [40]. Only the early diagnosis of neonatal meningitis allows correct therapy, reducing mortality and complications. This is the reason why, to define the appropriateness of empiric antimicrobials, we considered not only their in vitro activity but also their delivery through the blood-brain barrier when meningitis was confirmed (or not ruled out) through a lumbar puncture. Of clinical relevance, we found that the risk of sepsis-related mortality was 4- to 6-fold higher when empirical antimicrobials were inappropriate or had undetermined appropriateness, respectively. Interestingly, mortality was associated with the appropriateness of empirical antimicrobials, but not with their in vitro activity. The increased risk persisted in the multivariate model (after adjusting for confounding factors) only in cases of inappropriate antimicrobials. However, the risk of sepsis-related death was not significantly higher in cases where the appropriateness of empirical antimicrobials was undetermined (cases who did not receive a lumbar puncture and empirical antimicrobials did not cross the blood-brain barrier at high levels). It is possible that in those cases the lumbar puncture had been deferred because of the infant’s hemodynamic instability and increased disease severity at onset. Consequently, their risk of sepsis related mortality may have been biased. In addition, the association between increased sepsis-related mortality and inappropriate antimicrobial therapy persisted independently from the pathogen (Gram-negative vs. Gram-positive). This result appears relevant considering the higher (in vitro) resistance of Gram-negative pathogens, whose infections present with higher disease severity and fatal outcome [41]. Recent studies support an association between late-onset meningitis and intraventricular haemorrhage (grade 3 to 4) [42]. Notably, in our cohort, none of the infants who developed brain lesions had received appropriate empirical antimicrobials. Therefore, the appropriateness of empirical antimicrobials appears to have a profound impact on neonatal sepsis, by reducing both mortality and brain sequelae. This impact is probably greater than that of the timing and the in vitro activity of empirical antimicrobials. Thus, although the collection of cerebrospinal fluid before administering antibiotics is associated with a longer time to antibiotic administration [43], clinicians should reconsider the risk/benefit ratio of deferring (or not performing) the lumbar puncture in septic infants. When cerebrospinal fluid findings allow to rule out meningitis, empirical antimicrobials that do not necessarily pass the blood-brain barrier at high levels can be safely administered. In contrast, when (or until) the lumbar puncture cannot be performed, empirical treatment should include antimicrobials easily penetrating the blood-brain barrier. However, to avoid unnecessary antibiotics, we need objective criteria to timely identify septic high-risk very low birth weight infants who deserve this broad-spectrum antibiotic coverage highly active on the central nervous system.

Our study has several potential limitations. First, we included infants with multiple episodes of late-onset sepsis, perhaps increasing the risk of potential biases. However, all episodes were separated by a ≥7 days in-between period and caused by different pathogens, therefore, greatly minimizing the risk of considering the same episode multiple times. Secondly, although unlikely, the time to first antibiotics may have been underestimated, considering the retrospective design and the possibility of late registration of signs and symptoms of sepsis. However, as mentioned above, the electronic medical records available in our NICU make possible a sufficiently accurate assessment of the timing of symptom onset and antibiotic administration. Third, the effect of the appropriateness of empirical antimicrobials on neonatal outcome may be partly biased by a relatively high proportion of lumbar punctures not performed (or not sent to the laboratory for the analysis because of the inability to have a reliable result). This limitation is mainly related to the clinical instability of newborns and the technical difficulty of performing lumbar puncture in those of extreme gestational age, as also reported in cohort studies carried out in other countries [21,22]. Finally, the effect of time to first antibiotic could be also biased by the intrinsic virulence of the pathogen. Molecular testing was not routinely performed in our unit, and we could not assess the pathogen’s virulence, due to the retrospective design of our study. In the future, molecular markers of virulence should be assessed for each pathogen, since they can affect the efficacy of a timely administration of antibiotics

## 5. Conclusions

The assumption that a timely administration of antibiotics in the NICU can be life-saving is probably true, although confounding factors may obscure this effect. According to our results, the appropriateness of empirical antimicrobials, rather than their timing and in vitro activity, appears life- and brain-saving in septic very low birth weight infants. Cerebrospinal fluid findings obtained through a lumbar puncture at the onset of late-onset sepsis should guide the choice of empirical antimicrobials. Until meningitis has been ruled out, very low birth weight infants at high risk of sepsis-related mortality or brain sequelae should receive broad-spectrum antibiotics with high delivery through the blood-brain barrier.

## Figures and Tables

**Figure 1 microorganisms-11-00396-f001:**
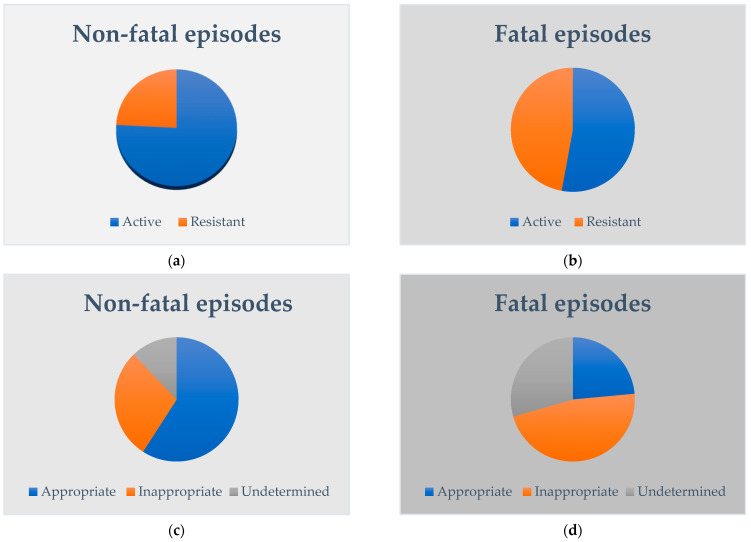
In vitro activity (**a**,**b**) and appropriateness (**c**,**d**) of empirical antimicrobials are compared between non-fatal and fatal episodes of late-onset sepsis among very low birth weight preterm infants.

**Figure 2 microorganisms-11-00396-f002:**
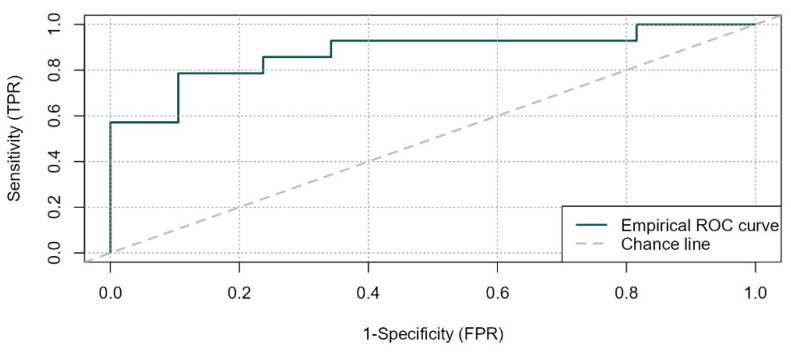
Empirical ROC curve (—) vs. Chance line (---) in multivariate analysis for sepsis-related mortality.

**Table 1 microorganisms-11-00396-t001:** Demographics of infants with late-onset sepsis episodes and characteristics of empiric antimicrobials. Categorical variables are reported as number and percentage. Continuous variables are reported as median and IQR.

Variables	All Episodes (*n* = 83)	Non-Fatal Episodes (*n* = 66)	Fatal Episodes (*n* = 17)	*p*
Male sex	42 (50.6)	32 (48.5)	10 (58.8)	0.59
Gestatational age, weeks	26.0 (25.0–27.0)	27.0 (25.0–28.0)	25.0 (24.0–26.0)	0.019
Birth weight, g	780.0 (654.0–993.8)	800.0 (697.0–1050.0)	690.0 (624.8–895.5)	0.050
Age at sepsis onset, days	18.0 (10.3–32.8)	22.0 (10.0–33.0)	14.0 (10.5–21.8)	0.281
Body weight at sepsis onset, days	1056.0 (766.8–1301.8)	1100.0 (885.0–1342.0)	760.0 (594.8–1078.3)	0.003
Gram-negative pathogen	50 (60.2)	36 (54.5)	14 (82.4)	0.051
Time to first antibiotics, hours	2 (0–6)	3 (0.5–10)	1 (0–2)	0.035
In vitro active empirical antimicrobials *	59 (71.1)	50 (75.8)	9 (52.9)	0.121
Appropriate empirical antimicrobials §	43 (51.8)	39 (59.1)	4 (23.5)	0.019
Appropriate empirical antimicrobials (cases with undetermined appropriateness excluded) ¶	43/70 (61.4)	39/58 (67.0)	4/12 (33.3)	0.048

* Active empirical antimicrobials: the pathogen was susceptible (in vitro) to at least one empirical antimicrobial (or each pathogen was susceptible to at least one antimicrobial in the case of polymicrobial infections). Inactive empirical antimicrobials: the pathogen was resistant to all empirical antimicrobials. § Appropriate empirical antimicrobials: the empirical antimicrobial was active (in vitro) against the pathogen and (i) meningitis was ruled out (through a lumbar puncture) or (ii) the antimicrobial ideally penetrated at high levels the blood-brain barrier. Inappropriate empirical antimicrobials: (i) inactive empirical antimicrobials or (ii) active empirical antimicrobials ideally did not pass the blood-brain barrier at high levels (e.g., an aminoglycoside) and lumbar puncture findings were consistent with meningitis. Undetermined appropriateness of empirical antimicrobials: active empirical antimicrobials ideally did not pass the blood-brain barrier at high levels (e.g., an aminoglycoside) and meningitis was not ruled out (no lumbar puncture was performed). ¶ Percentages are calculated by excluding cases for whom empirical antimicrobial effectiveness was undetermined (non-fatal episodes *n* = 8, fatal episodes *n* = 5).

**Table 2 microorganisms-11-00396-t002:** Uni- and multi-variate analysis of association between interventions and sepsis-related mortality in late-onset sepsis among preterm very low birth weight infants.

	Univariable Models		Multivariable Model	
	OR (95% CI)	*p*	aOR (95% CI)	*p*
Time to first antibiotics	0.85 (0.7–1.02)	0.087	0.90 (0.71–1.15)	0.408
Time to volume administration	0.92 (0.83–1.02)	0.129	0.89 (0.76–1.04)	0.130
Appropriate empirical antimicrobials	Reference		Reference	
Inappropriate empirical antimicrobials	4.11 (1.1–15.36)	0.036	10.25 (1.37–76.75)	0.023
Undetermined appropriateness of empirical antimicrobials	6.09 (1.33–27.83)	0.020	4.12 (0.42–40.68)	0.226
Gram-positive pathogen	Reference		Reference	
Gram-negative pathogen	3.89 (1.02–14.82)	0.047	9.85 (1.55–83.97)	0.036

**Table 3 microorganisms-11-00396-t003:** Variance inflation factor (*VIF*), degrees of freedom (*df*) and $VIF^{\frac{1}{2\times df}}$ for each term of the multivariable model.

	*VIF*	*df*	$VIF^{\frac{1}{2\times df}}$
Time to first antibiotics	1.033	1	1.017
Time to volume administration	1.348	1	1.161
Inappropriate empirical antimicrobials	1.665	2	1.136
Gram-negative pathogen	1.747	1	1.322

## Data Availability

The data that support the findings of this study are openly available (attachment to this submission).

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
