# Peer review of "Late-Onset Sepsis Mortality among Preterm Infants: Beyond Time to First Antibiotics"

_microorganisms, 2023, doi:10.3390/microorganisms11020396_

Round 1

Reviewer 1 Report

Treatment of sepsis remains a significant health challenge among preterm infants.  This contribution uses a retrospective approach to investigate the relative importance of timing of administration of antimicrobials, whether they are active against the identified microbe, and if they can penetrate to blood-brain barrier.  Hence, the authors address a critical issue in the NICU, and particularly for infants delivered at less than 32 weeks and at higher risk of sepsis and CNS involvement.  As with many (most) studies, additional questions arise.  The following comments are provided to guide the authors in considering relatively minor changes, including spelling and grammar.

1.       First line of intro: “al-“  instead of “alt-“

2.       Third line of intro: “presents”

3.       Next to last line of 2.2: death occurring within 7 days from the its onset

4.       “unsusceptible” can be replaced by “resistant” throughout the text.

5.       Inclusion criteria: VLBW infants can be confused with SGA infants born later in gestation.  It might be better to be consistent with using preterm infants delivered at less than 32 weeks of gestation.

6.       Exclusion criteria:

a.       “infeasible” to “not feasible”

b.       If it is difficult to distinguish between true infections and contamination for CoNS, why is this not true for other bacteria?

c.       Why would growth in a single culture be considered a contaminant?

d.       Fungal cultures were performed.  Were there attempts to identify viral infections?

7.         Outcomes:  What is the relationship between CNS infection and IVH?  Does CNS infection cause IVH or does IVH and major failure of the BBB allow for CNS infection.  A recent paper may be of interest (Zhou Q, Ong M, Lan M, Ye XY, Ting JY, Shah PS, Lee SK; Canadian Neonatal Network (CNN) Investigators. Decreasing Trend in Incidence of Late Onset Culture Positive Bloodstream Infections but Not Late Onset Meningitis in Preterm Infants <33 Weeks Gestation in Canadian Neonatal Intensive Care Unit. Neonatology. 2022;119(1):60-67).

8.         Results:

a.       When empirical antimicrobials were in vitro inactive, sepsis-related mortality was not significantly higher (OR: 2,778; IC 95% 0,918 – 8,397; P=0.070).”  Granted, this value is not <0.05, but is does suggest that even if in vitro inactive antimicrobial are administered, there is a “tendency” for mortality to be reduced.  Can the authors comment?

b.       Similarly, “no effect was detected when analysing how sepsis-related mortality varied by time to first antibiotics (OR: 0,849; IC 95% 0,703-1,024; p=0,087).  There does seem to be a “tendency” for timing to influence mortality.  Do the authors think this would become more or less significant with a larger sample size?  Might this “tendency” contribute to the differing opinions about timing?

c.       Was there concrodance between pathogens identified in the blood and CSF cultures?

d.       Positive CSF, negative blood – might these have been from infants that had 2nd or 3rd episodes after previous treatment, but with inappropriate antimicrobial? 

9.         Discussion:

a.       Second paragraph: Previous studies have “already” demonstrated

b.       Third paragraph: Most contraindications (“haemodynamic” instability

c.       Inter-

d.       “Interestingly, mortality was associated with the appropriateness of empirical antimicrobials, but not with their in vitro activity.” Can the authors provide speculation about this surprising finding that penetration of the BBB, without evidence of CNS infection or activity, is associated with lower mortality?

e.       in our cohort, none among of the infants

f.        Fourth paragraph: the electronical medical records

Author Response

We thank the reviewer for his/her precious work. Please find here a reply point by point.

Referee 1

  1. First line of intro: “al-“  instead of “alt-“. We are sorry but we cannot understand the exact correction to be done.
  2. Third line of intro: “presents” Done, thanks.
  3. Next to last line of 2.2: death occurring within 7 days from the its onset Done, thanks.
  4. “unsusceptible” can be replaced by “resistant” throughout the text. Done, thanks.
  5. Inclusion criteria: VLBW infants can be confused with SGA infants born later in gestation.  It might be better to be consistent with using preterm infants delivered at less than 32 weeks of gestation. That’s correct, we have modified as requested.
  6. Exclusion criteria:
  7. “infeasible” to “not feasible” Done, thanks.
  8. If it is difficult to distinguish between true infections and contamination for CoNS, why is this not true for other bacteria? The literature provides criteria to distinguish contamination from true infection for bacteria that belong to the skin microflora, including CoNS. In fact, these bacteria, colonizing the skin, are frequently reported as contaminants of blood cultures after a non-sterile collection. In contrast, in infants with symptoms of sepsis, isolation of pathogens from blood or cerebrospinal fluid usually identifies true infections.
  9. Why would growth in a single culture be considered a contaminant? According to some previous studies we considered such organisms as contaminants (see Stoll BJ,. Late-onset sepsis in very low birth weight neonates: the experience of the NICHD Neonatal Research Network. Pediatrics. 2002 Aug;110(2 Pt 1):285-91. doi: 10.1542/peds.110.2.285. PMID: 12165580)
  10. Fungal cultures were performed. Were there attempts to identify viral infections? We routinely look for viruses in infants with suspected sepsis. However, consistently with most of the literature regarding late-onset sepsis in very low birth weight and preterm infants, we included only infants suffering from bacterial or fungal infections identified from the laboratory database.
  11. Outcomes:  What is the relationship between CNS infection and IVH?  Does CNS infection cause IVH or does IVH and major failure of the BBB allow for CNS infection.  A recent paper may be of interest (Zhou Q, Ong M, Lan M, Ye XY, Ting JY, Shah PS, Lee SK; Canadian Neonatal Network (CNN) Investigators. Decreasing Trend in Incidence of Late Onset Culture Positive Bloodstream Infections but Not Late Onset Meningitis in Preterm Infants <33 Weeks Gestation in Canadian Neonatal Intensive Care Unit. Neonatology. 2022;119(1):60-67). We thank the referee for the suggestion, which we find interesting. We have added it to the references and mentioned it in the discussion.
  12. Results:
  13. When empirical antimicrobials were in vitro inactive, sepsis-related mortality was not significantly higher (OR: 2,778; IC 95% 0,918 – 8,397; P=0.070).”Granted, this value is not <0.05, but is does suggest that even if in vitro inactive antimicrobial are administered, there is a “tendency” for mortality to be reduced.  Can the authors comment?
  14. Similarly, “no effect was detected when analysing how sepsis-related mortality varied by time to first antibiotics (OR: 0,849; IC 95% 0,703-1,024; p=0,087).  There does seem to be a “tendency” for timing to influence mortality.  Do the authors think this would become more or less significant with a larger sample size?  Might this “tendency” contribute to the differing opinions about timing?

We agree with the referee: there might be a trend of association between death and both in vitro activity and time to first antibiotics, which may not reach significance because of the small sample size. However, with the same sample size, a strong(er) association emerged between death and appropriateness of antibiotic therapy (which is the core finding of the study). Furthermore, several high impact Journals (i.e. Pediatrics from USA) suggest to avoid the term "trend" when referring to p values near but not below 0.05. In those cases they suggest to simply report a difference and its C.I. with or without the p value.

  1. Was there concordance between pathogens identified in the blood and CSF cultures? Two neonates had 2 different pathogens yielded in the CSF; one neonate had 2 different pathogens yielded in blood. In those cases, we defined the appropriateness of antibiotics considering both pathogens.
  2. Positive CSF, negative blood – might these have been from infants that had 2nd or 3rd episodes after previous treatment, but with inappropriate antimicrobial? No, as stated in methods, we included only episodes separated by ≥ 7 days in-between period and caused by different pathogens. We specified it better in “Results”: All infants with positive cerebrospinal fluid culture and negative blood experienced the first episode of infection
  3. Discussion:
  4. Second paragraph: Previous studies have “already” demonstrated. Done, thanks.
  5. Third paragraph: Most contraindications (“haemodynamic” instability Done, thanks.
  6. Inter- We are sorry: we cannot understand. Could you please specify the requested correction?
  7. “Interestingly, mortality was associated with the appropriateness of empirical antimicrobials, but not with their in vitro activity.” Can the authors provide speculation about this surprising finding that penetration of the BBB, without evidence of CNS infection or activity, is associated with lower mortality? This is not correct, because the appropriateness of the empirical antimicrobials is defined considering both the evidence of CNS infection and the antimicrobials’ activity on CSF. The correct interpretation of appropriateness of an antibiotic is: i) the passage through the BBB in confirmed meningitis or ii) the in vitro activity when meningitis is ruled out.
  8. in our cohort, none among of the infants Done, thanks.
  9. Fourth paragraph: the electronical medical records We have corrected to “electronic medical records”. Thanks.

We warmly thank the referee for the valuable suggestions, and certainly, our manuscript has improved after the suggested changes.

Best regards

Alberto Berardi and co-workers

Reviewer 2 Report

This study aims to determine the relevance of distinguishing the relevance between early treatment with antibiotics without evidence or including a diagnosis that makes it possible to improve the efficiency of treatment, in the use of the type of antibiotics and the form of inoculation. This study is of intrinsic interest since it aims to reduce the mortality rate from bacterial sepsis that cause meningitis. However, there are a number of issues that need to be reviewed.

“In fact, during the first 90 days of life, the innate immune system (namely, phagocytes, natural-killer cells, antigen presenting cells, and the complement system) provide a defence against pathogens, which is weaker among very low birth weight infants”.

Must include a reference

“Decreased function of neutrophils and low concentrations of immunoglobulins increase the susceptibility of preterm infants to invasive infections, increasing their related morbidity and mortality”.

Must include a reference

“Hospitalized preterm infants are exposed to environmental organisms that might become pathogenic for their immature immune system. Contact with hospital staff, family members and contaminated equipment all rep- resent opportunities for pathogen exposure”.

Must be referenced properly

In the text you can read:

Notably, the association between survival and timely antibiotic administration may
be biased by multiple factors, including the timeliness of diagnosis, the implementation
of appropriate shock supportive measures, and most of all the effectiveness of the empiric
antimicrobial treatment [16].

It would be necessary to add the intrinsic susceptibility of the strains causing this aetiology and that it is convenient that they have been characterized molecularly to identify resistance/virulence mechanisms that additionally affect the times in which antibiotics take effect.

One of the elements in which this work focuses on determining the relevance of the time interval between the diagnosis of meningitis of bacterial origin and the application of antibiotic therapy, however the growth physiology of the strain in particular, is not contemplated. Understanding that the study does not have data on the specific growth rate (m) of the pathogen in vitro, it should be assumed that the time between diagnosis and administration of antibiotic therapy should be minimized, since the specific growth rate determines colonization. of the tissue and therefore affects the severity of sepsis (since it can determine the virulence capacity of a particular strain). This thesis is assumed in the conclusion of the document. I understand that it would be convenient to include in the discussion the relevance of considering the parameters of the growth physiology of the pathogens as a variable that affects the pathogenicity and virulence of the disease, since there are already works that delve in this direction.

https://doi.org/10.1016/j.ijfoodmicro.2021.109462

In the work, a susceptibility to the antibiotic is considered, however, there are no data on the pathogenicity of the strains associated with the specific growth rate, which is an element to consider in future studies of this type. Susceptibility is different depending on the physicochemical and physiological context in which the pathogen grows.

The study mentions that some of the cases had eukaryotic pathogens (of the genus Candida) as etiological agents, therefore, the effect of antibiotics for these particular cases will be very low. I understand that the results for these particular cases should be part of an additional protocol, and be treated independently or removed from the analysis.

In Table 2, in the Time to first antibiotics column and in the Time to volume administration row, the units are not listed. In the Appropriate empirical antimicrobials row, it is not clear why "reference" is used, it should be explained in the legend of the table, in the same way as for the Gram positive pathogen row.

One of the questions that, in my opinion, needs to be answered in the future refers to the effect of the presence of antibiotics in high concentrations in the brain of infants. Is there any study that specifically delves into the effect of the presence of these drugs? in the tissue once the blood brain barrie has been crossed? Is there any correlation between the molecular mechanism of action of the antibiotic on the physiology of the neurons? Is there any study that focuses on the adjuvant analysis of the potential cytotoxic effect of an antibiotic in high concentrations with a pathogenic process that results in sequelae in the central nervous system?. There are works that have already delved into this theoretical framework. https://doi.org/10.1111%2Fj.1365-2125.2011.03991.x

Referring to the thesis defended in this work, it is logical to infer that the efficacy of the antimicrobial on the physiology of bacterial growth (bacteriostasis / cell lysis) is more relevant to stop a process of bacterial sepsis compared to the administration of drugs with early antimicrobial activity, in this sense, it is necessary to clarify that although antibiotics have an effect despite the presence of resistance mechanisms (since the flow of ATP in the Bacteria Physiology is redirected to the resistance phenotype) , it is possible that this effect is not enough to stop the pathogenic process, since the mechanisms of resistance to antibiotics are able to counteract the high concentrations of antibiotics by various mechanisms, (modifying the permeability, regulating the expression of the pharmacological targets, etc. …).

In the text of the discussion it is said

Furthermore, in children with severe sepsis or septic shock (median 7 years of age), an escalating risk of mortality was observed with each hour delay from sepsis recognition to first effective antimicrobial administration, although this did not reach significance until 3 hours [10]

In this In this sense, it is worth noting that the increase in the risk of mortality cannot be ascribed solely to the time when the antibiotic was administered (which is very relevant because it imposes a limit on the distribution of the pathogen), the intrinsic pathogenicity of the pathogen must also be considered, which does not have to be associated with its susceptibility to the presence of antibiotics, but rather with virulence factors, evasion of pharmacological action, production of toxins or secondary metabolites that affect the affected tissue physiology in the sick person, conditioning the local pharmacokinetics and pharmacodynamics. This scenario should be considered in the future and promote the characterization by molecular markers of the most virulent pathogens, since they should be considered in a particular way in future studies. I understand that part of this reasoning should be in the discussion of the work

Author Response

We thank the reviewer for his/her precious work. Please find here a reply point by point.

Referee 2

“In fact, during the first 90 days of life, the innate immune system (namely, phagocytes, natural-killer cells, antigen presenting cells, and the complement system) provide a defence against pathogens, which is weaker among very low birth weight infants”. Must include a reference. You are right, done. Thanks.

“Decreased function of neutrophils and low concentrations of immunoglobulins increase the susceptibility of preterm infants to invasive infections, increasing their related morbidity and mortality”. Must include a reference You are right, done. Thanks.

“Hospitalized preterm infants are exposed to environmental organisms that might become pathogenic for their immature immune system. Contact with hospital staff, family members and contaminated equipment all rep- resent opportunities for pathogen exposure”. Must be referenced properly. You are right, done. Thanks.

In the text you can read:

“Notably, the association between survival and timely antibiotic administration may
be biased by multiple factors, including the timeliness of diagnosis, the implementation
of appropriate shock supportive measures, and most of all the effectiveness of the empiric
antimicrobial treatment” [16]. It would be necessary to add the intrinsic susceptibility of the strains causing this aetiology and that it is convenient that they have been characterized molecularly to identify resistance/virulence mechanisms that additionally affect the times in which antibiotics take effect. We thank the reviewer for his suggestion. Indeed, this is a highly discussed topic, unfortunately we have done a retrospective review of all cases of late-onset sepsis in preterm infants since the year 2010. Therefore, this information, not obtained routinely, is not available due to the retrospective collection of data.

One of the elements in which this work focuses on determining the relevance of the time interval between the diagnosis of meningitis of bacterial origin and the application of antibiotic therapy, however the growth physiology of the strain in particular, is not contemplated. Understanding that the study does not have data on the specific growth rate (m) of the pathogen in vitro, it should be assumed that the time between diagnosis and administration of antibiotic therapy should be minimized, since the specific growth rate determines colonization. of the tissue and therefore affects the severity of sepsis (since it can determine the virulence capacity of a particular strain). This thesis is assumed in the conclusion of the document. I understand that it would be convenient to include in the discussion the relevance of considering the parameters of the growth physiology of the pathogens as a variable that affects the pathogenicity and virulence of the disease, since there are already works that delve in this direction. We agree, we added a sentence in the Discussion (limitations) of the study. Thanks.

https://doi.org/10.1016/j.ijfoodmicro.2021.109462

In the work, a susceptibility to the antibiotic is considered, however, there are no data on the pathogenicity of the strains associated with the specific growth rate, which is an element to consider in future studies of this type. Susceptibility is different depending on the physicochemical and physiological context in which the pathogen grows. We agree, and we added a sentence in the Discussion (limitations) of the study

The study mentions that some of the cases had eukaryotic pathogens (of the genus Candida) as etiological agents, therefore, the effect of antibiotics for these particular cases will be very low. I understand that the results for these particular cases should be part of an additional protocol, and be treated independently or removed from the analysis. In the case of fungal infections, we considered all antibiotic therapy as inappropriate, we considered appropriate only the active antifungal therapy. A few studies on the timing of neonatal antibiotics (which we cited in the references) also included fungal infections, because in preterm infants they can have a high morbidity and mortality rate (Baczynski, J Pediatr 2021)

In Table 2, in the Time to first antibiotics column and in the Time to volume administration row, the units are not listed. In the Appropriate empirical antimicrobials row, it is not clear why "reference" is used, it should be explained in the legend of the table, in the same way as for the Gram positive pathogen row. Please let us specify  that we did not include units as we believe they could be misleading, since the table show to confidence intervals and odds ratio.

One of the questions that, in my opinion, needs to be answered in the future refers to the effect of the presence of antibiotics in high concentrations in the brain of infants. Is there any study that specifically delves into the effect of the presence of these drugs? in the tissue once the blood brain barrie has been crossed? Is there any correlation between the molecular mechanism of action of the antibiotic on the physiology of the neurons? Is there any study that focuses on the adjuvant analysis of the potential cytotoxic effect of an antibiotic in high concentrations with a pathogenic process that results in sequelae in the central nervous system?. There are works that have already delved into this theoretical framework. https://doi.org/10.1111%2Fj.1365-2125.2011.03991.x In the newborn, pharmacokinetics studies of antibiotics are very poor, and many drugs are off label, so unfortunately we cannot answer the question posed by you.

Referring to the thesis defended in this work, it is logical to infer that the efficacy of the antimicrobial on the physiology of bacterial growth (bacteriostasis / cell lysis) is more relevant to stop a process of bacterial sepsis compared to the administration of drugs with early antimicrobial activity, in this sense, it is necessary to clarify that although antibiotics have an effect despite the presence of resistance mechanisms (since the flow of ATP in the Bacteria Physiology is redirected to the resistance phenotype) , it is possible that this effect is not enough to stop the pathogenic process, since the mechanisms of resistance to antibiotics are able to counteract the high concentrations of antibiotics by various mechanisms, (modifying the permeability, regulating the expression of the pharmacological targets, etc. …).

In the text of the discussion it is said

Furthermore, in children with severe sepsis or septic shock (median 7 years of age), an escalating risk of mortality was observed with each hour delay from sepsis recognition to first effective antimicrobial administration, although this did not reach significance until 3 hours [10]

In this In this sense, it is worth noting that the increase in the risk of mortality cannot be ascribed solely to the time when the antibiotic was administered (which is very relevant because it imposes a limit on the distribution of the pathogen), the intrinsic pathogenicity of the pathogen must also be considered, which does not have to be associated with its susceptibility to the presence of antibiotics, but rather with virulence factors, evasion of pharmacological action, production of toxins or secondary metabolites that affect the affected tissue physiology in the sick person, conditioning the local pharmacokinetics and pharmacodynamics. This scenario should be considered in the future and promote the characterization by molecular markers of the most virulent pathogens, since they should be considered in a particular way in future studies. I understand that part of this reasoning should be in the discussion of the work. Very interesting comment, we added this sentence at the end of the Discussion: Molecular markers of virulence of each pathogen should be assessed in future studies, since they can affect the efficacy of a timely administration of antibiotics.

We warmly thank the referee for the valuable suggestions, and certainly, our manuscript has improved after the suggested changes.

Best regards

Alberto Berardi and Co-workers
